# Real-Life Outcomes of Adjuvant Targeted Therapy and Anti-PD1 Agents in Stage III/IV Resected Melanoma

**DOI:** 10.3390/cancers16173095

**Published:** 2024-09-06

**Authors:** Gabriele Roccuzzo, Paolo Fava, Chiara Astrua, Matteo Giovanni Brizio, Giovanni Cavaliere, Eleonora Bongiovanni, Umberto Santaniello, Giulia Carpentieri, Luca Cangiolosi, Camilla Brondino, Valentina Pala, Simone Ribero, Pietro Quaglino

**Affiliations:** Section of Dermatology, Department of Medical Sciences, University of Turin, 10126 Turin, Torino, Italy; fava_paolo@yahoo.it (P.F.); castrua@cittadellasalute.to.it (C.A.); matteobrizio@virgilio.it (M.G.B.); g.cavali_dr@yahoo.it (G.C.); eleonora.bongiovanni@unito.it (E.B.); umberto.santaniello@edu.unito.it (U.S.); giulia.carpentier@edu.unito.it (G.C.); luca.cangialosi@edu.unito.it (L.C.); camilla.brondino@edu.unito.it (C.B.); valentinapala@live.it (V.P.); simone.ribero@unito.it (S.R.); pietro.quaglino@unito.it (P.Q.)

**Keywords:** melanoma, adjuvant, BRAF, dabrafenib, nivolumab, pembrolizumab

## Abstract

**Simple Summary:**

Adjuvant therapy with targeted therapy or immunotherapy has become the standard of care for fully resected stage III–IV melanoma. In this scenario, real-world data are needed to relate the actual effectiveness and safety of these regimens with the evidence provided in the clinical trials. This study provides clinicians and researchers with the results of an Italian single-center real-world experience on the use of adjuvant therapy in resected stage III–IV melanoma patients. Our findings confirm the real-world effectiveness and safety of adjuvant regimens, yet underscores the need for further research to explore biomarker-based predictors for relapse and to assess the translation of improved relapse-free survival into long-term overall survival benefit.

**Abstract:**

This study was carried out at the Dermatologic Clinic of the University of Turin, Italy, to assess the effectiveness and safety of adjuvant therapy in patients who received either targeted therapy (TT: dabrafenib + trametinib) or immunotherapy (IT: nivolumab or pembrolizumab) for up to 12 months. A total of 163 patients participated, including 147 with stage III and 19 with stage IV with no evidence of disease. The primary outcomes were relapse-free survival (RFS), distant metastasis-free survival (DMFS), and overall survival (OS). At 48 months, both TT and IT approaches yielded comparable outcomes in terms of RFS (55.6–55.4%, *p* = 0.532), DMFS (58.2–59.8%, *p* = 0.761), and OS (62.4–69.5%, *p* = 0.889). Whilst temporary therapy suspension was more common among TT-treated patients compared to IT-treated individuals, therapy discontinuation due to adverse events occurred at comparable rates in both groups. Predictors of relapse included mitoses, lymphovascular invasion, ulceration, and positive sentinel lymph nodes. Overall, the proportion of BRAF-mutated patients receiving IT stood at 7.4%, lower than what was observed in clinical trials.

## 1. Introduction

Melanoma is an aggressive form of skin cancer with a steadily increasing incidence [1,2]. In the last decade, the introduction of systemic therapy in the adjuvant setting has lowered the risk of recurrence in stage III and IV disease-free patients. Stage III melanoma-specific survival (MSS) ranges from 88% (stage IIIA) to 24% (stage IIID) at 10 years, with adjuvant regimens currently based on a 12-month cycle of either an immune-checkpoint inhibitor or a targeted therapy combination (TT) [3,4]. As for immunotherapy (IT), the anti-PD-1 (programmed death-1) agents nivolumab and pembrolizumab gained approval regardless of the patient’s BRAF mutation status, whilst the combination of the BRAF inhibitor dabrafenib and the MEK inhibitor trametinib can only be offered to BRAF-mutated patients [4,5]. The efficacy of adjuvant therapy was proven by the pivotal phase III randomized clinical trials Keynote-054 (pembrolizumab vs. placebo), CheckMate-238 (nivolumab vs. ipilimumab), and COMBI-AD (dabrafenib-trametinib vs. placebo) [6,7,8,9,10,11]. Significant improvement in terms of RFS (relapse-free survival) was reported with nivolumab (5-year RFS 50% vs. 39% for ipilimumab), pembrolizumab (5-year RFS 55.4% vs. 38.3% for placebo), and the combination of dabrafenib and trametinib (5-year RFS 52% vs. 36% for placebo) [6,7,8,9,10,11]. Despite the improvement in prognosis determined by adjuvant therapy, recurrence remains frequent and the impact on long-term overall survival still needs to be elucidated [12]. This retrospective study aims to evaluate the real-world outcomes of melanoma patients treated with adjuvant therapy at a specialized Italian tertiary referral center. It also seeks to assess the safety profile of these treatments and analyze the observed recurrence patterns.

## 2. Materials and Methods

A retrospective series of melanoma patients treated with adjuvant therapy at the Dermatology Clinic of the Turin University Hospital, Italy, between September 2017 and April 2023 was collected. All patient information was sourced from the hospital’s database and subsequently archived within an internal computerized database. Patient inclusion criteria were: age ≥ 18 years, histologically confirmed diagnosis of melanoma, tumor stage defined as fully resected stage IIIA-D or IV-NED (AJCC 8th edition) [13], and absence of evidence of distant metastasis before the start of adjuvant therapy according to total-body computed tomography scans. Lymphadenectomy following a positive sentinel lymph node biopsy (SLNB) was not mandatory, but it was discussed on a case-by-case basis within the tumor board, considering the patient’s clinical circumstances and the timing of the procedure (i.e., prior to or after the results of the MSLT2 trial) [14,15]. The type of adjuvant regimen was chosen in a multidisciplinary setting considering BRAF status and comorbidities (e.g., active autoimmune disease). Adjuvant therapy lasted until the completion of the one-year cycle unless there was disease progression or unacceptable toxicity. Study endpoints were the following: relapse-free survival (RFS), as the time from the start of therapy to the date of the first recurrence or death from any cause; distant metastasis-free survival (DMFS), as the time from the start of therapy to the date of the development of distant metastases or death from any cause; and overall survival (OS), as the time from the start of therapy until death. For patients alive without disease recurrence nor metastasis development data were censored on the date of last patient contact. Descriptive statistics were used for patient and tumor characteristics. Mann–Whitney, Chi-squared with Yates corrections, and Fisher’s exact tests were used to analyze continuous and paired nominal data. To address potential confounding factors due to the lack of randomization, Cox regression models were employed to adjust for imbalances between treatment groups. Such analyses were limited to independent variables with available data in over 75% of the patients, as per common practice. Diagnostics through variance inflation factor (VIF) were used to rule out multicollinearity among independent variables. Model fitness was evaluated according to McFadden’s formula and the Hosmer and Lemeshow test. The proportional hazards assumption on the basis of Schoenfeld residuals was tested after fitting the Cox models [16]. Survival curves were generated based on the Kaplan–Meier method and analyzed through a Log-rank test. A *p*-value of ≤0.05 was considered statistically significant. All statistical analyses were performed using Stata/SE.v.18 Software (StataCorp, College Station, TX, USA).

## 3. Results

A total of 163 patients were included. The baseline characteristics of the patients are summarized in Table 1. The IT-treated and TT-treated groups showed comparable baseline characteristics in terms of demographics and histological features, yet there was a slightly higher prevalence of stage IIIA in the TT group (19.5% vs. 9.9%, *p* = 0.022) and of occult and mucosal melanomas in the IT group (18.5 vs. 3.6%). Since patients with IV-NED disease did not undergo SLNB and all received adjuvant nivolumab according to prescribing policies, this staging procedure was more common in the TT group. Overall, a total of 85 patients (52.1%) underwent lymphadenectomy, either with (n = 61 patients) or without (n = 24 patients) prior positive SLNB. Overall, lymph node involvement at histological evaluation was confirmed in 49.4% of dissected patients. In terms of molecular characteristics, 86 patients (52.7%) displayed BRAF mutation (100% in the TT group vs. 7.4% in the IT group, *p* < 0.001), while 33 patients exhibited NRAS mutant melanoma (39.5% in the IT group vs. 1.2% in the TT group, *p* < 0.001).

### 3.1. Effectiveness and Safety of Adjuvant Treatment

In total, 82 patients (50.3%) received treatment with TT dabrafenib + trametinib, while 81 patients (49.7%) underwent adjuvant therapy with anti-PD-1 (63 patients with nivolumab, 18 patients with pembrolizumab). Out of 86 patients diagnosed with BRAF-mutant melanoma (52.8%), 81 were treated with TT (94.2%). The cumulative RFS rate over 48 months was 54.9% (95% CI, 45.0–63.7), specifically 56.7% (95% CI, 45.9–66.3) for stage-III and 35.7% (95% CI, 13.7–58.7) for IV-NED (*p* = 0.007). Survival rates for different stage-IIIs were recorded as 78.0% (95% CI, 53.7–90.6) for IIIA, 66.0% (95% CI, 47.2% to 79.5%) for IIIB, and 47.6% (95% CI, 30.6–62.7) for IIIC (not reached for IIID) (Figure 1a). This breakdown showed 55.6% (95% CI, 42.0% to 67.2%) in the TT group and 55.4% (95% CI, 41.9% to 67.0%) in the IT group, with no statistically significant differences between treatment categories (*p* = 0.532) or among the three drug types used (*p* = 0.754) (Figure 1b). 

For the 48-month DMFS rate, it was 58.4% (95% CI, 48.0% to 67.3%) for the entire population, specifically 60.1% (95% CI, 49.1% to 70.2%) for stage III and 35.7% (95% CI, 13.7% to 58.7%) for IV-NED (*p* = 0.001). DMFS for different stage IIIs was recorded as 78.0% (95% CI, 53.7–90.6) for IIIA, 69.3% (95% CI, 49.7% to 82.4%) for IIIB, and 52.5% (95% CI, 33.8–68.0) for IIIC (not reached for IIID) (Figure 2a). Within the TT group, the rate was 58.2% (95% CI, 44.1% to 69.9%), while in the IT group, it reached 59.8% (95% CI, 45.5% to 71.5%). Similar to RFS, differences in DMFS between the two treatment categories (*p* = 0.761) and the three drug types did not show any statistical significance (*p* = 0.666) (Figure 2b).

Lastly, the overall 48-month OS rate was calculated as 66.5% (95% CI, 55.5% to 75.3%), specifically 67.1% (95% CI, 55.1% to 76.6%) for stage III and 57.5% (95% CI, 28.3% to 78.5%) for IV-NED (*p* = 0.105). This further broke down into 62.4% (95% CI, 44.6% to 75.9%) in the TT group and 69.5% (95% CI, 55.0% to 80.1%) in the IT group. As for different stage IIIs, the following OS rates were recorded: 83.2% (95% CI, 55.2–94.5) for IIIA, 80.1% (95% CI, 60.2% to 90.8%) for IIIB, and 61.4% (95% CI, 42.8–75.5) for IIIC (not reached for IIID) (Figure 3a). Consistently, there were no statistically significant differences between the two treatment categories (*p* = 0.889) or the three drug types (*p* = 0.989) (Figure 3b). 

Overall, 123 patients (75.7%) completed the one-year cycle of adjuvant treatment, whereas 17 patients (10.4%) interrupted the therapy beforehand due to disease progression and 21 patients (12.9%) due to adverse event. While temporary therapy suspension was more common in TT-treated patients compared to IT-treated ones (68.3% vs. 13.6%, *p* < 0.001), therapy discontinuation secondary to adverse events was comparable in both groups (11.1% vs. 14.8%, respectively, *p* = 0.464). In total, 38 patients (23.3%) died, 19 of them received TT, and 19 IT (16 nivolumab, 3 pembrolizumab). 

### 3.2. Patterns of Recurrence and Predictors of Outcomes

At 48 months, disease recurrence was observed in 57 patients (35.0%), of whom 29 (50.9%) had received TT and 28 (49.1%) IT (22 nivolumab and 6 pembrolizumab). As for the site of recurrence, 8 patients (14.0%) had a loco-regional recurrence, with only local skin and/or lymph nodes involved, whereas 38 patients (66.6%) developed only distant metastases and 11 (19.3%) both local and distant metastasis. As for the timing, most TT-treated patients relapsed after the end of the adjuvant cycle (22 patients—75.9%), whereas only seven patients (24.1%) relapsed during adjuvant treatment. Conversely, most IT-treated patients recurred during the treatment (19 patients—67.9%), whereas only nine patients (32.1%) relapsed after the end of adjuvant therapy. In terms of recurrence sites, the most common localizations were the lung (n = 20), brain (n = 20), regional lymph nodes (n = 18), and skin (n = 16), followed by the liver (n = 6), gastrointestinal tract (n = 6), skeletal apparatus (n = 4), kidney (n = 2), and adrenal glands (n = 2). In terms of different therapeutic regimens, a higher incidence of brain relapse was noted in TT-treated patients (48.3% vs. 21.4%, *p* = 0.034), while no significant differences were observed for other sites (Table 2). 

In the context of first-line therapy following relapse, immunotherapy was administered to 30 patients with the following distribution: 6 (20%) received a combination of nivolumab and ipilimumab, 16 (53.3%) were treated with anti-PD1 single agent, and 6 (20%) received ipilimumab as a single agent. In contrast, first-line targeted therapy was prescribed for 12 patients. The incorporation of these treatments was complemented by additional modalities, including 11 cases of surgery (36.7%), 13 cases of stereotactic radiation therapy (43.3%), and 5 (16.7%) instances of electrochemotherapy (ECT). In two patients for whom other therapeutic options were not suitable, chemotherapy with temozolomide was administered. The univariate Cox regression analysis for RFS revealed significant associations with various predictors, such as age, stage IIID, stage IV, Breslow thickness, ulceration, number of mitoses, lymphovascular invasion, and number of positive sentinel lymph nodes. For DMFS, univariate analysis confirmed significant associations with stage IIID, stage IV-NED, and ulceration. Predictors for OS in the univariate analysis included age, stage IIID, ulceration, lymphovascular invasion, number of positive sentinel lymph nodes, and distant relapse. In the multivariate analysis for RFS, significance was maintained in the context of the combination of number of mitoses–lymphovascular invasion and ulceration–number of positive sentinel lymph nodes (Table 3).

## 4. Discussion

This study presents the findings derived from a population of melanoma patients undergoing adjuvant therapy in a real-life setting, aiming to evaluate the implications of commonly employed outcomes in clinical trials, such as RFS, DMFS, and OS. Contrary to historical data, current insights indicate a significant improvement in RFS for resectable stage III/IV-NED melanoma with the introduction of adjuvant therapy, albeit with variations in efficacy and benefits across cohorts [6,7,8,9,10,11]. In this intricate landscape, real-world inquiries remain indispensable for comprehending potential similarities and disparities compared to efficacy rates observed in clinical trials. For example, the CheckMate-238 trial enrolled fully resected patients in stages IIIB, IIIC, and IV-NED [6], while the Keynote-054 trial included stages IIIA, IIIB, and IIIC according to the AJCC classification 7th Edition [8]. In contrast, our retrospective study encompassed the entire population eligible for adjuvant therapy, ranging from fully resected melanoma patients in stages IIIA-D to IV-NED, in accordance with Italian national guidelines and following the AJCC 8th Edition [15]. Regarding RFS, our real-life study results align with the CheckMate-238 and COMBI-AD trials, reporting RFS rates of 51.7% and 55% at 4 years [7,10,17]. Similarly, our study documented RFS rates of 55.4% in the IT group and 55.6% in the TT group, mirroring the trial results [7,10,17]. Our findings also correspond with the Keynote-054 update, displaying a 3.5-year DMFS rate of 65.3% (61% at 42 months in our study) and the CheckMate-238 trial at 4 years (60% in the trial vs. 59.8% in our study) [8]. Interestingly, the slightly higher RFS rates observed can be partially attributed to the presence of stage IIIA patients receiving nivolumab, while data for stage IIIB (66.0% vs. 66.0%) and IIIC (47.6% vs. 47%) mirror the results of the Checkmate-238 trial following revision with the 8th AJCC Edition [18]. Unfortunately, the absence of data at the 48-month mark for stage IIID patients results from the constrained sample size within this subgroup, a limitation noted in other real-life studies addressing this subgroup class introduced with the new AJCC Edition [19,20]. Thus far, in terms of OS, only ipilimumab has demonstrated a significant extension compared to placebo [18,21]. Nivolumab’s superiority over ipilimumab was established concerning RFS and the safety profile, despite similar DMFS trends [18,21]. For BRAF-mutant patients treated with dabrafenib and trametinib, despite initially showing a 3-year OS advantage over placebo (86% vs. 77%) in the preliminary interim analysis, a statistically significant difference did not meet pre-defined criteria in the clinical trial [17]. Notably, our study does not reveal substantial differences between IT- and TT-treated patients, yet discloses some variations in OS rates at 4 years (69.5% in the IT group compared to 77.9% in the CheckMate trial), plausibly due to the reduction in the number of patients at risk from year 3 (n = 74) to year 4 (n = 22) due to sample size limitations, impacting the estimation of 48-month OS probabilities. Significantly, within the 38 recorded deaths, 6 were unrelated to melanoma (with a mean age of 72.0 years). This aspect could potentially have led to a marginal overestimation of the death rate, highlighting it as a limitation associated with the sample size. All these elements depict the heterogeneous landscape of real-world melanoma practice, with substantial differences compared to the controlled trial populations. In our experience, both TT and IT groups presented similar baseline characteristics, except for a higher percentage of mucosal melanoma in the IT group (due to a lower incidence of BRAF mutations in this subset) and a higher number of SNLB performed in the BRAF-positive group, as the IT-treated group included stage IV-NED patients typically not requiring this procedure. Disease recurrence manifested in 35.0% of patients, with similar rates observed following both TT and immunotherapy IT regimens, predominantly occurring at distant sites, as documented in other studies [22,23]. Notably, 35.1% of patients experiencing recurrence exhibited brain metastasis—a heightened occurrence compared to the pre-TT/IT era [24]. This increase is likely attributed to shifts in clinical practices, where routine surveillance brain imaging has become standard, leading to a higher detection rate of asymptomatic brain metastases [22,23,25,26]. This underscores a notable shift in the paradigm of relapse detection compared to the pre-therapy era, when relapses were primarily identified clinically [24]. Also, distinct patterns of relapse were observed, as patients receiving IT tended to experience higher rates of relapse during their treatment, whereas those on TT more frequently relapsed after treatment had concluded. Despite the absence of known molecular mechanisms directly connecting BRAF/MAPK pathway upregulation to increased formation of brain metastases, the possibility of crosstalk with the PI3K pathway has been described [27,28]. Notably, inhibiting the MAPK pathway through BRAF and MEK inhibition may result in resistance, as the parallel PI3K/AKT pathway is upregulated [27]. The recent COMBI-MB clinical study suggests that the brain is a prominent site of treatment failure after BRAF inhibition [29]. In our investigation, we identified various clinical and histological factors associated with the relapse event. For instance, our findings reaffirm the independent prognostic importance of mitotic rate and ulceration in predicting relapse risk, emphasizing their potential in refining risk stratification [30,31]. Nevertheless, it underscores the ongoing necessity to uncover new predictors for potential recurrence, such as circulating tumor DNA, to enhance the ability to predict patients susceptible to relapse [32,33]. Furthermore, completion lymphadenectomy’s impact on the risk of relapse (HR 0.95, 95% CI 0.58–1.57, *p* = 0.856) and death (HR 0.91, 95% CI 0.48–1.74, *p* = 0.781) was found to be insignificant, aligning with existing literature. This reinforces the notion that its role in cases of positive SLNB should be considered solely in select cases [14,34,35]. Regarding safety, 68.3% on TT and 13.6% on IT temporarily halted treatment due to adverse events, and these figures align with prior reports [20,22,23]. Interestingly, comparable rates of therapy discontinuation secondary to adverse events in both groups were observed, in contrast with the higher discontinuation rates originally reported for TT (26% in COMBI-AD) compared to IT (7.7% in CheckMate-238, 13.8% in Keynote-054) [7,8,10]. In our real-life experience, the numerous cases of pyrexia resulting from treatment with TT were generally mild and temporary, leading to permanent therapy discontinuation only in a minority of cases (i.e., 7%). This evidence reinforces that both treatment regimens demonstrate a favorable safety and tolerance profile in real-life settings. Although our study had limitations stemming from its retrospective design, including missing data from cases initially diagnosed in other facilities but later referred to our referral center for adjuvant therapy initiation, our research supports the real-world effectiveness of both adjuvant treatment approaches. Overall, the proportion of BRAF-mutated patients receiving IT stood at 7.4%, notably lower than what was observed in clinical trials (41% in CheckMate-237 and 35% in Keynote-054). This discrepancy reflects a prevailing inclination in Italy to administer adjuvant TT to this subset of patients [20]. While in other retrospective investigations, BRAF-mutated patients undergoing IT exhibited a lower 24-month RFS compared to those treated with TT, our four-year follow-up did not corroborate this trend, despite confirming variations in recurrence kinetics, particularly in the short term [36]. In these regards, future research endeavors are imperative to include the identification of biomarker-based predictors for relapse and to further evaluate the translation of improved RFS into OS benefits [37,38,39,40].

## 5. Conclusions

These findings confirm that both targeted therapy and immunotherapy regimens are effective and maintain safe profiles. However, approximately half of the patients eventually experience a relapse, highlighting the ongoing challenge of predicting which patients are at risk. To enhance patient outcomes, future research must focus on identifying reliable clinical, histological, and molecular predictors of relapse, as well as predictors of sustained response. Additionally, determining the most effective therapy sequencing for patients who relapse is essential. Addressing these challenges is crucial for advancing personalized treatment approaches and improving survival rates in melanoma patients.

## Figures and Tables

**Figure 1 cancers-16-03095-f001:**
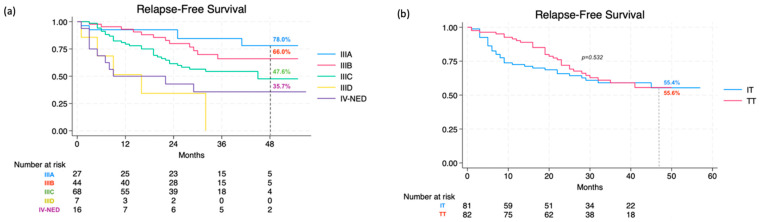
Relapse-free survival at 48 months according to stage (**a**) and therapy (**b**).

**Figure 2 cancers-16-03095-f002:**
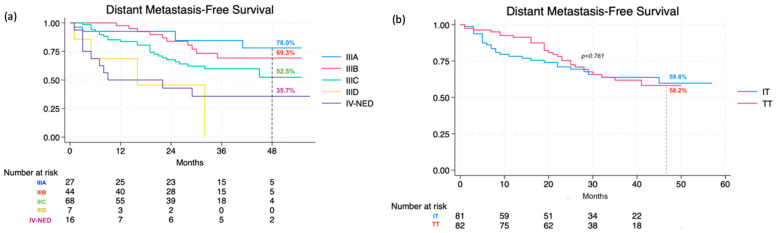
Distant metastasis-free survival at 48 months according to stage (**a**) and therapy (**b**).

**Figure 3 cancers-16-03095-f003:**
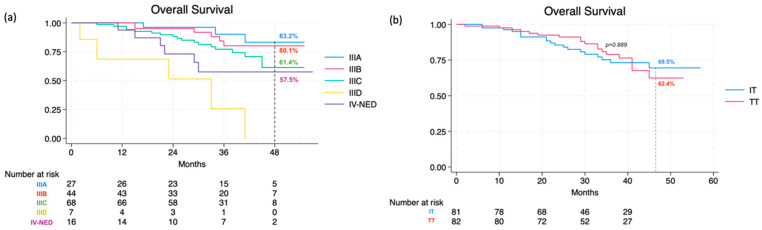
Overall survival at 48 months according to stage (**a**) and therapy (**b**).

**Table 1 cancers-16-03095-t001:** Patients’ characteristics according to clinical and histological features *.

	Global (n = 163)	Immunotherapy (n = 81)	Nivolumab (n = 63)	Pembrolizumab (n = 18)	Dabrafenib + Trametinib (n = 82)	*p*-Value
Median age—years (range)	55 (18–89)	60 (20–85)	60 (20–85)	60.5 (33–78)	51 (18–89)	0.089
Male sex—no (%)	83 (50.9)	40 (49.4)	29 (46)	11 (61.1)	43 (52.4)	0.696
Female sex—no (%)	80 (49.1)	41 (50.6)	34 (54)	7 (38.9)	39 (47.6)	0.696
Stage III—no (%)	147 (90.2)	65 (80.2)	47 (74.6)	18 (100)	82 (100)	-
IIIA	27 (16.6)	8 (9.9)	4 (6.4)	4 (22.2)	19 (19.5)	**0.022**
IIIB	44 (26.9)	25 (23.5)	15 (23.8)	10 (55.5)	19 (23.2)	0.713
IIIC	70 (43.0)	29 (35.8)	25 (39.7)	3 (16.7)	40 (48.8)	0.054
IIID	7 (4.3)	3 (3.7)	2 (1.6)	1 (5.6)	4 (4.9)	0.414
Stage IV-NED—no (%)	16 (9.8)	16 (19.8)	16 (25.4)	0 (0.0)	0 (0.0)	-
Brain	4 (25)	4 (25)	4 (25)	0 (0.0)	0 (0.0)	-
Lung	6 (37.5)	6 (37.5)	6 (37.5)	0 (0.0)	0 (0.0)	-
Liver	1 (6.3)	1 (6.3)	1 (6.3)	0 (0.0)	0 (0.0)	-
Ileum	2 (12.5)	2 (12.5)	2 (12.5)	0 (0.0)	0 (0.0)	-
Pancreas	1 (6.3)	1 (6.3)	1 (6.3)	0 (0.0)	0 (0.0)	-
Lymph nodes	2 (12.5)	2 (12.5)	2 (12.5)	0 (0.0)	0 (0.0)	-
In transit metastases—no (%)	27 (16.6)	16 (19.8)	10 (15.9)	6 (33.3)	11 (13.4)	0.276
1 metastasis	21 (77.8)	13 (81.3)	8 (80)	5 (83.3)	8 (72.7)	0.230
≥2 metastases	6 (22.2)	3 (18.7)	2 (20)	1 (16.7)	3 (27.3)	0.496
Site of primary melanoma—no (%)						
Back	49 (30.1)	25 (30.9)	19 (30.2)	6 (33.3)	24 (29.3)	0.824
Lower limbs	34 (20.1)	13 (12.3)	12 (19.0)	1 (5.5)	21 (25.6)	0.133
Abdomen	17 (10.4)	6 (7.4)	5 (7.9)	1 (5.5)	11 (13.4)	0.219
Upper limbs	17 (10.4)	10 (12.3)	10 (15.9)	0 (0.0)	7 (8.5)	0.426
Head–neck	15 (9.2)	8 (9.8)	5 (7.9)	3 (16.6)	7 (8.5)	0.767
Thorax	13 (7.9)	4 (4.9)	1 (1.6)	3 (16.6)	9 (10.9)	0.154
Occult melanoma	11 (6.7)	9 (11.1)	8 (12.7)	1 (5.5)	2 (2.4)	**0.028**
Mucosal	7 (4.3)	6 (7.4)	3 (4.7)	3 (16.6)	1 (1.2)	**0.050**
Histology—no (%)						
SSM	65 (39.9)	28 (34.6)	21 (33.3)	7 (38.9)	37 (45.1)	0.168
Nodular	31 (19)	13 (16)	11 (17.5)	2 (11.1)	18 (22)	0.337
LMM	2 (1.2)	2 (2.5)	1 (1.6)	1 (5.6)	0 (0.0)	0.245
Mucosal	7 (4.3)	6 (7.4)	3 (4.8)	3 (16.7)	1 (1.2)	0.050
Occult melanoma	11 (6.7)	9 (11.1)	8 (12.7)	1 (5.6)	2 (2.4)	0.028
Not otherwise specified	29 (25)	15 (32.1)	14 (36.5)	1 (16.7)	16 (23.2)	0.202
Breslow thickness—no (%)						
<1 mm	11 (6.7)	5 (6.1)	4 (6.4)	1 (5.6)	6 (7.3)	0.770
1–2 mm	38 (23.3)	16 (19.8)	7 (11.1)	9 (50)	22 (26.8)	0.285
2.1–4 mm	43 (26.4)	20 (24.7)	19 (30.1)	1 (5.5)	23 (28)	0.626
>4 mm	46 (28.2)	23 (28.4)	20 (31.8)	3 (16.7)	23 (28)	0.960
Not otherwise specified	25 (15.3)	17 (21)	13 (20.6)	4 (22.2)	8 (9.8)	0.076
Lymphovascular invasion—no (%)	21 (12.9)	9 (11.1)	7 (11.1)	2 (11.1)	12 (14.6)	0.502
Ulceration—no (%)	77 (47.2)	36 (44.4)	27 (42.9)	9 (50)	41 (50)	0.477
Mitotic index—median (range)	5 (0–27)	5 (0–27)	5 (1–17)	4 (0–27)	4 (0–20)	0.850
BRAF mutated, no (%)	88 (53.9)	6 (7.4)	5 (8.0)	1 (5.5)	82 (100)	**<0.001**
NRAS mutated—no (%)	33 (20.2)	32 (39.5)	23 (36.5)	9 (50)	1 (1.2)	**<0.001**
Lymphatic disease—no (%)						
Macroscopic involvement	35 (21.5)	20 (24.7)	17 (27)	3 (16.7)	15 (18.3)	0.319
Microscopic involvement	99 (60.7)	36 (44.4)	26 (41.3)	10 (55.6)	63 (76.8)	**<0.001**
Absent	29 (17.8)	25 (30.9)	20 (31.7)	5 (27.8)	4 (4.9)	**<0.001**
SLNB—no (%)						
Performed	123 (75.5)	54 (66.7)	42 (66.7)	12 (66.7)	69 (84.1)	**0.015**
Not performed	40 (24.5)	27 (33.3)	21 (33.3)	6 (33.3)	13 (15.9)	-
SLNB positive	103 (83.7)	39 (72.2)	29 (69)	10 (83.3)	64 (92.8)	**0.004**
SLNB negative	20 (16.3)	15 (27.8)	13 (31)	2 (16.7)	5 (7.2)	-
1 pos. lymph node	73 (59.3)	21 (38.9)	15 (35.7)	6 (50)	52 (75.4)	**<0.001**
2 pos. lymph nodes	23 (18.7)	14 (25.9)	10 (23.8)	4 (33.3)	9 (13)	**0.003**
3–4 pos. lymph nodes	7 (5.7)	4 (7.5)	4 (9.5)	0 (0.0)	3 (4.3)	0.698
Size of metastasis in sentinel lymph node—median (range)	1.75 (0.10–18.0)	1.8 (0.3–18.0)	1.5 (0.3–8.0)	4.0 (0.3–18.0)	1.7 (0.10–12.0)	0.850
Lymphadenectomy– no (%)						
Performed	85 (52.1)	39 (48.1)	32 (50.8)	7 (38.9)	46 (56.1)	0.309
Not performed	78 (47.9)	42 (51.9)	31 (49.2)	11 (61.1)	36 (43.9)	0.309
0 pos. lymph nodes	43 (50.6)	18 (46.2)	15 (46.9)	3 (42.9)	25 (54.3)	0.451
1 pos. lymph node	24 (28.2)	14 (35.9)	10 (31.3)	4 (57.1)	10 (21.7)	0.148
2 pos. lymph nodes	8 (9.4)	2 (5.1)	2 (6.3)	0 (0.0)	6 (13)	0.279
≥3 pos. lymph nodes	10 (11.9)	5 (13.0)	5 (15.5)	(0.0)	5 (10.9)	1

* The descriptive comparison refers to the immunotherapy vs. targeted therapy groups of patients. Statistically significant figures are depicted in bold.

**Table 2 cancers-16-03095-t002:** Patterns of recurrence and therapy discontinuation.

	Global	Immunotherapy (n = 81)	Targeted Therapy (n = 82)	*p*-Value
Adjuvant treatment completed—no (%)	123 (75.7)	58 (47.1)	65 (52.9)	0.255
Patients recurred—no (%)	57 (35.0)	28 (34.6)	29 (35.4%)	0.699
Recurred ON therapy—no (%)	26 (45.6)	19 (67.9)	7 (24.1)	**0.001**
Recurred OFF therapy—no (%)	31 (54.4)	9 (32.1)	22 (75.9)	**0.001**
Site of recurrence—no				
- Regional lymph nodes	18	10	8	0.509
- Brain	20	6	14	**0.034**
- Lungs	20	13	7	0.078
- Skin	16	8	8	0.934
- Liver	6	5	1	0.076
- Gastrointestinal	6	4	2	0.363
- Kidney	2	1	1	0.980
- Adrenal glands	2	1	1	0.980
Dose interruption due to adverse events *—no (%)	67 (41.1)	11 (13.6)	56 (68.3)	**<0.001**
Therapy discontinuation due to adverse event—no (%)	21 (12.9)	12 (14.8)	9 (11.1)	0.464
End of therapy due to disease progression—no (%)	17 (10.4)	10 (58.8)	7 (41.2)	0.426
Death—no (%)	38 (23.3)	19 (23.5)	19 (23.2)	0.973

* Most common AEs in IT-treated: endocrinal disorders, skin reactions, GI distress; most common AEs in TT: pyrexia, fatigue, nausea. Statistically significant figures are depicted in bold.

**Table 3 cancers-16-03095-t003:** Significant clinical and histological predictors of relapse and death.

Univariate					Multivariate ^a^		
Outcome	Predictor	HR	95% CI	*p*-Value	HR	95% CI	*p*-Value
RFS	Age	1.02	1.01–1.04	0.035	-		
	Stage IIIA	0.35	0.14–0.86	0.023	-		
	Stage IIID	3.75	1.50–9.41	0.005	-		
	Stage IV-NED	2.42	1.23–4.77	0.011	-		
	Breslow thickness	1.05	1.01–1.10	0.026	-		
	Number of mitoses	1.06	1.01–1.11	0.023	1.07	1.01–1.13	0.028
	Lymphovascular Invasion	2.30	1.14–4.63	0.020	2.37	1.17–4.79	0.017
	Ulceration	2.29	1.25–4.20	0.007	2.79	1.39–5.63	0.004
	No. of positive sentinel LNs	1.46	1.04–2.06	0.032	1.44	1.04–2.01	0.027
DMFS	Stage IIIA	0.40	0.16–0.99	0.049	-		
	Stage IIID	4.32	1.34–13.87	0.014	-		
	Stage IV-NED	2.85	1.19–6.87	0.019	-		
	Ulceration	2.57	1.31–4.99	0.005	-		
OS	Age	1.04	1.01–1.06	0.004	1.04	1.01–1.08	0.011
	Stage IIID	6.89	2.67–17.81	<0.001	-		
	Ulceration	2.80	1.26–6.22	0.011	-		
	Lymphovascular invasion	2.67	1.16–6.13	0.021	2.53	1.09–5.87	0.031
	No. of positive sentinel LNs	1.60	1.06–2.42	0.026	-		
	Skin relapse	0.21	0.07–0.61	<0.001	-		
	Distant Relapse	43.72	10.51–181.68	<0.001	40.40	9.68–168.57	<0.001

^a^ The number of variables included in the multivariate analysis was considered in relation to the number of events to avoid overfitting.

## Data Availability

Data are available upon reasonable request to the corresponding author.

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
