# Peer review of "Real-Life Outcomes of Adjuvant Targeted Therapy and Anti-PD1 Agents in Stage III/IV Resected Melanoma"

_cancers, 2024, doi:10.3390/cancers16173095_

Round 1
Reviewer 1 Report
Comments and Suggestions for Authors
This study a real-world study on the use of adjuvant therapy in resected stage III-IV melanoma patients are described for a relatively low number of patients. The objective of the study was to compare the effects and safety of two (targeted therapy and immuntherapy) treatments.
The authors compare evaluate the real-life outcomes of adjuvant targeted therapy (BRAF inhibitors) and anti-PD1 2 agents in stage III/IV resected melanoma in different melanoma groups. A retrospective survey was conducted on a minimally homogenized patient group (patients diagnosed with stage IIIA-D or IV-NED melanoma who did not have visible metastasis before adjuvant therapy). My opinion is that the conditions for a cohort study or clinical trial do not really met. The first problem I have is that the number of patients within each group are uneven (IIIA=27; IIIb=44; IIIC=70 and IIID=7; in transit metastases - no meta=21; meta=4; BRAF mutated n=88; NRAS mutated n=33). I don’t know how appropriate to compare the treatment of BRAF mutated (TARGETED THERAPY) and the treatment of patients without BRAF mutation even those belong to the same stage, there are no control group in the study.
The authors found that “temporary therapy suspension was more common among TT-treated patients compared to IT-treated individuals” is it really appropriate to compare this parameter between the two groups even the patients have different mutation status, therefore the biological behaviour of tumors are different?
Notes:
Abstract: It is hard to read the abstract, because it contains a large number of abbreviations. From the abstract it is unclear how many patients with BRAF mutant melanoma received only immunotherapy, so the statement in the last sentence is not entirely relevant. Even though the Simple Summary is well phrased, the Abstract should be rewritten to make it easier to read.
Introduction, contains all the necessary information, and the references are up-to-date.
Results: Overall, the results have very low statistical power. The low statistical power can be attributed to the low number of cases, especially in some subgroups. I am not sure how acceptable such a low case number is, regardless, this is a human epidemiological survey where selecting an adequate sample size is a basic requirement. In the case of retrospective data collection based on medical records, it is always a concern how reliable and complete the recorded data are.
When interpreting the results, it may be problematic to determine how comparable the groups are. Specifically, whether the difference (in survival) is due to the differing effectiveness of the two adjuvant methods or because the initial patient groups (before treatment) were already different from each other. The allocation of patients to one treatment or the other was not random (in fact, quite the opposite): “The type of adjuvant regimen was chosen in a multidisciplinary setting considering BRAF status and comorbidities.”
For me, the question is whether it makes sense to conduct analyses comparing the differences between the two (TT and IT) interventions (despite the fact that most of the results obtained are not significant).
… “In terms of different therapeutic regimens, a higher incidence of brain relapse was noted in the TT cohort (48.3% vs 21.4%, p=0.034), while no significant differences were observed for other sites." The result could be explained 'in theory' by the group allocation.
In general, the authors consistently mention cohorts, but this was not a cohort study (and it is difficult to imagine this based on the available data). The research question is confusing: '...to evaluate the real-world effectiveness and safety of adjuvant therapy in patients treated with either targeted therapy (TT: dabrafenib + trametinib) or immunotherapy (IT: nivolumab or pembrolizumab). Due to the low statistical power and lack of randomization, 'effectiveness' cannot be reliably evaluated between the two groups, and if there is no control group, then we have nothing to compare it to.
In summary I agree with the authors that in order to enhance patient outcomes, future research must focus on identifying reliable clinical, histological, and molecular predictors of relapse, as well as predictors of sustained response. I agree with the formulation of the conclusion.
Author Response
Dear Editor,
I am pleased that our paper has been taken into consideration by your honored journal.
I thank the reviewers for the much-appreciated comments which have been extremely helpful in furtherly improving our draft.
Below I will give a step-by-step response to the reviewers' suggestions.
-This study a real-world study on the use of adjuvant therapy in resected stage III-IV melanoma patients are described for a relatively low number of patients. The objective of the study was to compare the effects and safety of two (targeted therapy and immuntherapy) treatments. The authors compare evaluate the real-life outcomes of adjuvant targeted therapy (BRAF inhibitors) and anti-PD1 2 agents in stage III/IV resected melanoma in different melanoma groups. A retrospective survey was conducted on a minimally homogenized patient group (patients diagnosed with stage IIIA-D or IV-NED melanoma who did not have visible metastasis before adjuvant therapy). My opinion is that the conditions for a cohort study or clinical trial do not really met. The first problem I have is that the number of patients within each group are uneven (IIIA=27; IIIb=44; IIIC=70 and IIID=7; in transit metastases - no meta=21; meta=4; BRAF mutated n=88; NRAS mutated n=33). I don’t know how appropriate to compare the treatment of BRAF mutated (TARGETED THERAPY) and the treatment of patients without BRAF mutation even those belong to the same stage, there are no control group in the study.
Reply: We appreciate the reviewer’s comment. We acknowledge the intrinsic limitations of our study related to the sample size, as pointed out in the discussion section: "Unfortunately, the absence of data at the 48-month mark for stage IIID patients results from the constrained sample size within this subgroup, a limitation noted in other real-life studies addressing this subgroup class introduced with the new AJCC Edition" (....) "This aspect could potentially have led to a marginal overestimation of the death rate, highlighting it as a limitation associated with the sample size" (...) "discloses some variations in OS rates at 4 years (69.5% in the IT-group compared to 77.9% in the CheckMate trial), plausibly due to the reduction in the number of patients at risk from year 3 (n=74) to year 4 (n=22) due to sample size limitations, impacting the estimation of 48-month OS probabilities." Of note, as a single-center study, the collection of data from 163 patients is a substantial number, especially when compared to other recently published studies from our country (e.g., DOI: 10.1002/ijc.34462). Regarding the "minimally homogenized" population, it is important to note that the stage 3 disease was evenly distributed across the various therapy groups, without significant differences except for IIIA (addressed in the paper) and with corresponding figures for IIIB, despite all IV-NED patients being included in the immunotherapy group. This finding arises from the fact that in Italy, nivolumab is the only approved treatment for patients with resected stage IV melanoma, and adjuvant therapy with targeted therapies cannot be prescribed (as outlined in the EADO guidelines as well). Of note, in the pivotal trial (https://doi.org/10.1016/j.annonc.2020.08.1200) these patients depicted relapse trends in line with stage IIIC patients, and therefore are usually presented together with resected stage III across several real-world studies (e.g. DOI: 10.1111/jdv.18779). It is also important to clarify that the primary aim of our study was not to assess the superiority or inferiority of targeted therapies in BRAF mutant/WT patients, but rather to evaluate the real-life outcomes in terms of relapse rates and the potential differences in relapse sites. The absence of a control group is a limitation intrinsic to the study design, as all patients received therapy as per standard of care.
-The authors found that “temporary therapy suspension was more common among TT-treated patients compared to IT-treated individuals” is it really appropriate to compare this parameter between the two groups even the patients have different mutation status, therefore the biological behaviour of tumors are different?
Reply: We thank the reviewer for the interesting question. Indeed, we believe that these data are noteworthy to report as they highlight what the clinicians and patients have experienced in this real-life scenario. As pointed out, these data are somehow different from the clinical trials and this finding needs to be mentioned.
- Abstract: It is hard to read the abstract, because it contains a large number of abbreviations. From the abstract it is unclear how many patients with BRAF mutant melanoma received only immunotherapy, so the statement in the last sentence is not entirely relevant. Even though the Simple Summary is well phrased, the Abstract should be rewritten to make it easier to read.
Reply: We thank the reviewer for the suggestion. We rewrote the abstract accordingly. Please see revised draft.
-Introduction, contains all the necessary information, and the references are up-to-date.
Reply: We thank the reviewer for the positive feedback.
-Results: Overall, the results have very low statistical power. The low statistical power can be attributed to the low number of cases, especially in some subgroups. I am not sure how acceptable such a low case number is, regardless, this is a human epidemiological survey where selecting an adequate sample size is a basic requirement. In the case of retrospective data collection based on medical records, it is always a concern how reliable and complete the recorded data are.
Reply: We thank the reviewer for the comment. Indeed the sample size is a limitation of our study. We addressed it in the discussion. As for the specific power analysis, it was not calculated in a formal way as per common practice in observational studies (https://doi.org/10.1016/j.jclinepi.2021.08.028).
-When interpreting the results, it may be problematic to determine how comparable the groups are. Specifically, whether the difference (in survival) is due to the differing effectiveness of the two adjuvant methods or because the initial patient groups (before treatment) were already different from each other. The allocation of patients to one treatment or the other was not random (in fact, quite the opposite): “The type of adjuvant regimen was chosen in a multidisciplinary setting considering BRAF status and comorbidities.” For me, the question is whether it makes sense to conduct analyses comparing the differences between the two (TT and IT) interventions (despite the fact that most of the results obtained are not significant). “In terms of different therapeutic regimens, a higher incidence of brain relapse was noted in the TT cohort (48.3% vs 21.4%, p=0.034), while no significant differences were observed for other sites." The result could be explained 'in theory' by the group allocation.
Reply: We appreciate the reviewer’s suggestion. The prescription process adhered to the following considerations: BRAF status (if positive, both treatment regimens were available as per guidelines) and the presence of comorbidities that contraindicated either specific therapy (e.g., active autoimmune disease for immunotherapy, reduced ejection fraction for targeted therapy). These decisions were independent of baseline tumor characteristics, reflecting common practice in real-world settings where patients often present with comorbidities not typically accounted for in highly controlled clinical trials. We specified this info in the Methods section. Regarding the brain relapses, these events were not predictable based on baseline tumor features. Moreover, all patients had a negative baseline CT/PET-CT scan and MRI, making this an independent finding.
-In general, the authors consistently mention cohorts, but this was not a cohort study (and it is difficult to imagine this based on the available data). The research question is confusing: '...to evaluate the real-world effectiveness and safety of adjuvant therapy in patients treated with either targeted therapy (TT: dabrafenib + trametinib) or immunotherapy (IT: nivolumab or pembrolizumab). Due to the low statistical power and lack of randomization, 'effectiveness' cannot be reliably evaluated between the two groups, and if there is no control group, then we have nothing to compare it to.
Reply: We thank the reviewer for the comment. We changed the wording (e.g., group of patients, population of patients) to the adhere to the suggested definition. Please see revised draft. As for effectiveness, we opted for such definition as “efficacy” is usually mentioned in pivotal clinical trials. The study limitations in terms of sample size have been addressed as previously mentioned.
-In summary I agree with the authors that in order to enhance patient outcomes, future research must focus on identifying reliable clinical, histological, and molecular predictors of relapse, as well as predictors of sustained response. I agree with the formulation of the conclusion.
-We thank the reviewer for the positive feedback.
Reviewer 2 Report
Comments and Suggestions for Authors
Interesting findings by the authors. I have a suggestion for the authors, besides using nivolumab, the authors could also target IDO1 using inhibitors to prevent the relapse of the cancer and improve relapse free survival in patients. Such a study has been published (Nat Med 27, 2212–2223 (2021). https://doi.org/10.1038/s41591-021-01544-x) which the authors must discuss. The authors must also discuss the roles of IDO1, PD-L1 in melanoma (PMID: 29331888, 10.1136/jitc-2023-006755, 10.1126/scitranslmed.3006504 etc). IDO1 is a very important heme protein having very important role in immune suppression in cancers, the heme insertion mechanism into IDO1 was recently discovered (PMID: 35051612) and the role of nitric oxide in regulating the heme levels and ultimately the activity of IDO1 was recently published too (PMID: 37116709). These works if discussed in the discussion section would greatly enrich the paper and provide a future direction as well, which would help prevent relapse in melanoma patients. Thank you.
Author Response
Dear Editor,
I am pleased that our paper has been taken into consideration by your honored journal.
I thank the reviewers for the much-appreciated comments which have been extremely helpful in furtherly improving our draft.
Below I will give a step-by-step response to the reviewers’s suggestions.
-Interesting findings by the authors. I have a suggestion for the authors, besides using nivolumab, the authors could also target IDO1 using inhibitors to prevent the relapse of the cancer and improve relapse free survival in patients. Such a study has been published (Nat Med 27, 2212–2223 (2021). https://doi.org/10.1038/s41591-021-01544-x) which the authors must discuss. The authors must also discuss the roles of IDO1, PD-L1 in melanoma (PMID: 29331888, 10.1136/jitc-2023-006755, 10.1126/scitranslmed.3006504 etc). IDO1 is a very important heme protein having very important role in immune suppression in cancers, the heme insertion mechanism into IDO1 was recently discovered (PMID: 35051612) and the role of nitric oxide in regulating the heme levels and ultimately the activity of IDO1 was recently published too (PMID: 37116709). These works if discussed in the discussion section would greatly enrich the paper and provide a future direction as well, which would help prevent relapse in melanoma patients. Thank you.
Reply: We thank the reviewer for the comment. Indeed, IDO1-targeting represents a potential important tool for the future, among others. However, as its use has not been approved in Italy, our data focused only on the currently available therapeutic options. We included the suggested citations on the role in TT/IT regimens and the clinical trial for reference in the “future biomarkers” section of the discussion. Thanks again for the positive feedback on our paper.
Round 2
Reviewer 1 Report
Comments and Suggestions for Authors
The authors answered all the questions and comments and rewrote the manuscript, which definitelly improved.
I accept the answers.
Minor comment: the authors should check the genes name all should be italic : like BRAF mutation etc.
Author Response
We thank the reviewer for the positive feedback. We checked the typing of the genes as suggested. Thanks!